# Identification and Characterization of Novel ACE Inhibitory and Antioxidant Peptides from *Sardina pilchardus* Hydrolysate

**DOI:** 10.3390/foods12112216

**Published:** 2023-05-31

**Authors:** Mingyang Shao, Haixing Wu, Bohui Wang, Xuan Zhang, Xia Gao, Mengqi Jiang, Ruiheng Su, Xuanri Shen

**Affiliations:** 1School of Food Science and Engineering, Hainan University, Haikou 570228, China; smytun919@gmail.com (M.S.); wangbohui1012@163.com (B.W.); zhangxuan@hainanu.edu.cn (X.Z.); gaoxia20202021@163.com (X.G.); 20095135210016@hainanu.edu.cn (M.J.); 18715068801@163.com (R.S.); 2Key Laboratory of Food Nutrition, Functional Food of Hainan Province, Haikou 570228, China; 3Hainan Engineering Research Center, Aquatic Resources Efficient Utilization in South China Sea, Hainan University, Haikou 570228, China

**Keywords:** *Sardina pilchardus*, ACE inhibitory activity, antioxidant activity, LC-MS/MS, molecular docking

## Abstract

*Sardina pilchardus* is a valuable source of bioactive peptides with potential applications in functional foods. In this study, we investigated the angiotensin-converting enzyme (ACE) inhibitory activity of Sardina pilchardus protein hydrolysate (SPH) produced using dispase and alkaline protease. Our results showed that the low molecular mass fractions (<3 kDa) obtained through ultrafiltration exhibited more effective ACE inhibition, as indicated by screening with ACE inhibitory activity. We further identified the low molecular mass fractions (<3 kDa) using an LC-MS/MS rapid screening strategy. A total of 37 peptides with potential ACE inhibitory activity were identified based on high biological activity scores, non-toxicity, good solubility, and novelty. Molecular docking was used to screen for peptides with ACE inhibitory activity, resulting in the identification of 11 peptides with higher -CDOCKER ENERGY and -CDOCKER INTERACTION ENERGY scores than lisinopril. The sequences FIGR, FILR, FQRL, FRAL, KFL, and KLF were obtained by synthesizing and validating these 11 peptides in vitro, all of which had ACE inhibitory activity, as well as zinc-chelating capacity. All six peptides were found to bind to the three active pockets (S1, S2, and S1’) of ACE during molecular docking, indicating that their inhibition patterns were competitive. Further analysis of the structural characteristics of these peptides indicated that all six peptides contain phenylalanine, which suggests that they may possess antioxidant activities. After experimental verification, it was found that all six of these peptides have antioxidant activities, and we also found that the SPH and ultrafiltration fractions of SPH had antioxidant activities. These findings suggest that *Sardina pilchardus* may be a potential source of natural antioxidants and ACE inhibitors for the development of functional foods, and using LC-MS/MS in combination with an online database and molecular docking represents a promising, effective, and accurate approach for the discovery of novel ACE inhibitory peptides.

## 1. Introduction

Hypertension is a major global health concern and is the leading cause of cardiovascular disease and premature death [1]. With an increase in unhealthy lifestyles and an aging society, it has become imperative for the medical community to develop effective measures to prevent and control hypertension. The angiotensin-converting enzyme (ACE) plays a crucial role in regulating blood pressure in the human body. By converting angiotensin I to angiotensin II, ACE causes vasoconstriction, aldosterone secretion, and sodium and water retention, leading to an increase in blood pressure [2]. Moreover, ACE degrades diastolic kinins, which act as vasodilators in the trypsin-kinin system, further increasing blood pressure [3]. Therefore, controlling ACE activity is a key factor in controlling blood pressure. There is also a close relationship between antioxidant activity and ACE inhibitory activity. In many cases, there is a positive correlation between ACE inhibitory activity and antioxidant activity, meaning that peptides with ACE inhibitory activity often exhibit antioxidant activity as well [4,5,6]. This may be determined by their amino acid composition and sequence. For example, peptide sequences containing aromatic amino acids (phenylalanine, tyrosine, and tryptophan) are believed to have higher ACE inhibitory activity and also exhibit good antioxidant activity [7].

While traditional antihypertensive medications can effectively lower blood pressure, they may cause various side effects [8]. To address this issue, researchers have explored the potential of bioactive peptides derived from food proteins, which have shown significant hypotensive effects and appear to be safe with minimal side effects compared to conventional medications [9,10]. However, the complex composition and wide molecular mass dispersion of protein hydrolysates make it challenging to identify ACE inhibitory peptides [11,12]. Traditional separation and purification techniques, such as gel filtration chromatography and ion exchange chromatography, can be costly and often yield suboptimal results. Therefore, bioinformatics has become a promising solution to this problem. By integrating electronic tools and databases such as ACEpepDB (https://webs.iiitd.edu.in/raghava/ahtpdb/) (accessed on 28 March 2023), BIOPEP (http://www.uwm.edu.pl/biochemia/index.php/en/biopep) (accessed on 28 March 2023), and EROP-Moscow (http://erop.inbi.ras.ru) (accessed on 28 March 2023), the screening of bioactive peptides by LC-MS/MS combined with online databases has emerged as a more efficient, accurate, and less labor-intensive method. Silicon analysis has also been utilized to successfully identify some ACE inhibitory peptides. These advancements hold promise for more effective and targeted approaches to controlling hypertension [13,14].

Molecular docking, a computational biology technique that combines structural biology and bioinformatics, has revolutionized the peptide screening process by allowing for efficient and accurate identification of putative bioactive compounds. This method circumvents the tedious and time-consuming process of peptide purification and biological evaluation. Furthermore, virtual screening can help to elucidate the binding interactions between ligands and receptors. Tran et al. applied a novel three-step virtual screening approach to identify 16 ACE inhibitory peptides with IC50 values ranging from 5.6 to 274.4 μM [15]. Similarly, Yu et al. utilized in silico techniques to identify three ACE inhibitory peptides (EGF, HGR, and VDF) with IC50 values of 474.65, 106.21, and 439.27 μM, respectively [16]. These discoveries offer promising therapeutic agents for controlling blood pressure by inhibiting ACE activity.

The Chinese province of Hainan boasts a wealth of *Sardina pilchardus* (Walbaum, 1792), a fish species known for its rapid growth, high fertility, low cost, and high concentration of essential amino acids beneficial for human health [17]. While it has been reported that *S. pilchardus* protein includes ACE inhibitory peptides, there has been a lack of high-throughput screening and identification techniques employed thus far [18,19,20,21,22,23,24,25,26]. In light of this, our goal is to utilize LC-MS/MS in conjunction with an online database and a rapid screening method for molecular docking to identify potential ACE inhibitory peptides from *S. pilchardus* hydrolysates. In this study, both in silico and experimental analyses were performed to investigate the bioactivity of hydrolysates. In the experimental analysis, we prepared and separated the hydrolysates by ultrafiltration and identified peptides through LC-MS/MS. The bioactivity of the identified peptides was screened using in silico analysis, including various online tools and software to predict bioactivity, toxicity, and water solubility. Subsequently, molecular docking was performed to predict the ACE inhibition activity of the peptides. The peptides that showed potential to inhibit ACE were then subjected to in vitro activity testing, and the mechanism of effective ACE inhibitory peptides was further investigated.

## 2. Materials and Methods

### 2.1. Materials

Raw sardines (*S. pilchardus*) were purchased from Xiangtai Fisheries Co., Ltd. (Haikou, Hainan, China). ACE (from rabbit lung) and N-Hippuryl-His-Leu (HHL) were purchased from Sigma Chemicals Co., Ltd. (St. Louis, MO, USA). Dispase (enzyme activity 200 U/mg) was purchased from Adamas Chemicals Co., Ltd. (Shanghai, China). Alkaline protease (enzyme activity 200 U/mg) and borate buffer solution (BBS, pH8.4, containing 0.1 M H_3_BO_3_ and 0.025 M Na_2_B_4_O_7_) were purchased from Shanghai Yuanye Biotechnology Co., Ltd. (Shanghai, China). Trifluoroacetic acid (TFA), formic acid (FA), and acetonitrile (ACN) were obtained from Thermo Fisher Scientific Co., Ltd. (Waltham, MA, USA). The materials used for HPLC analysis were of HPLC grade, and all other reagents were of analytical grade. Artificial gastric fluid (10 g/L pepsin) and artificial intestinal fluid (10 g/L pancreatic enzyme) were purchased from Guangzhou Testing Technology Co., Ltd. (Guangzhou, China).

### 2.2. Preparation and Isolation of S. pilchardus Protein Hydrolysate

#### 2.2.1. Preparation of *S. pilchardus* Protein Hydrolysate

The *S. pilchardus* were grated into a crude powder with a mill and then defatted three times with ethanol (1:2.5, g/mL). The defatted powder was dried at 55 °C for 12 h to obtain *S. pilchardus* protein powder (SPP).

Enzymatic hydrolysis of SPP was carried out by adding 10 U/mg of dispase to a 4% (*w*/*w*) SPP suspension in deionized water. The shaker temperature, speed, and enzymatic digestion time were adjusted to 50 °C, 200 rpm, and 3 h, respectively. The pH was adjusted to 10 with 1 mol/L NaOH, and alkaline protease was added at a 10 U/mg enzyme–substrate ratio. The enzymatic digestion was completed by increasing the temperature to 60 °C and keeping it for 3 h. Then, the samples were boiled in water for 10 min and centrifuged at 10,000× *g*, 25 °C for 10 min. The supernatant is SPH. The SPH was freeze-dried and stored at −20 °C until use.

#### 2.2.2. Ultrafiltration Separation

To obtain distinct fractions of SPH, 20 mg/mL SPH solution was prepared by dissolving SPH powder in deionized water. The solution was then filtered through ultrafiltration cups with 3 and 5 kDa ultrafiltration membranes to obtain four fractions: <3 kDa, 3–5 kDa, <5 kDa, and >5 kDa. The ACE inhibitory activity of each fraction was measured as described in Section 2.6, and the fraction with the highest ACE inhibitory activity was selected for further analysis. This fraction was lyophilized and stored at −20 °C.

### 2.3. Peptide Sequence Analysis Based on LC-MS/MS

#### 2.3.1. Sample Preparation

The extracted peptides were lyophilized to almost dryness after being reduced by 10 mM DTT at 56 °C for 1 h and alkylated by 50 mM iodoacetamide at 25 °C in the dark for 40 min. Before LC-MS/MS analysis, the peptides were resuspended in 20 μL of 0.1% formic acid.

#### 2.3.2. Nano LC-MS/MS Analysis

EASY-nLC 1200 UPLC system (Thermo Fisher Scientific, USA) and Q Exactive Hybrid Quadrupole-Orbitrap Mass Spectrometer (Thermo Fisher Scientific, USA) were used to determine the amino acid sequence of peptides. The peptide elution was performed using 0.1% FA in water and 0.1% FA/80% ACN in water as mobile phase A and B, respectively, on an Acclaim PepMap RPLC analysis column (C18, 50 μm × 150 mm, 2 μm, 100 Å). The gradient elution conditions were as follows: 0–2 min, 4–8% B; 2–45 min, 8–28% B; 45–55 min, 28–40% B; 55–56 min, 40–95% B; 56–66 min, 95% B, respectively. The injection volume and elution flow rate were 5 μL and 600 nL/min, respectively.

The mass was obtained in a data-dependent manner with the Q Exactive Hybrid Quadrupole-Orbitrap Mass Spectrometer and automatically switched between MS and MS/MS scans. When MS spectra had a resolution of 70,000, the AGC target was 3 × 10^6^ ions, and the scan range was 100–1500 m/z. The activation type was HCD (higher energy C trap dissociation). The normalized coll energy was set to 28.0%. The resolution of MS/MS spectra was 17,500, and the AGC target was 1 × 10^5^ ions.

Based on the species of the samples, the raw MS files were examined and searched against the target protein database using Byonic. The parameters were set as follows: the protein modifications were carbamidomethylation (C) (fixed) and oxidation (M) (variable); the enzyme specificity was set to Non specific; the maximum missed cleavages were set to 3; the precursor ion mass tolerance was set to 20 ppm, and MS/MS tolerance was 0.02 Da. For the downstream protein identification analysis, only identified peptides with a high degree of confidence were chosen.

### 2.4. Physicochemical Properties and Toxicity Prediction of Peptides in SPH (<3 kDa)

The expression of bioactive peptides in vivo depends on solubility [27]; another crucial indicator of bioactive peptides is toxicity. To be ready for the subsequent step of docking with ACE crystal structure molecules, the critical indicators of bioactivity, toxicity, and solubility were used to evaluate the ACE inhibitory activity. Biological activity was predicted by Peptide Ranker (http://distilldeep.ucd.ie/PeptideRanker/) (accessed on 28 March 2023), and Toxicity tests were performed by ToxinPred (http://crdd.osdd.net/raghava/toxinpred/multi_submit.php) (accessed on 28 March 2023). Solubility prediction was performed with PepCalc (http://pepcalc.com) (accessed on 28 March 2023). The peptides with bioactivity scores greater than 0.8, non-toxic, and water-soluble were further screened. At the same time, the selected peptides have not been identified from the AHTPDB database (http://crdd.osdd.net/raghava/ahtpdb/pepsearch.php) (accessed on 5 April 2023), such as ACEpepDB (https://cftri.res.inpepdb/) (accessed on 5 April 2023), BIOPEP (http://www.uwm.edu.pl/biochemia/index.php/pl/biopep) (accessed on 5 April 2023), and EROP-Moscow (http://erop.inbi.ras.ru/index.html) (accessed on 5 April 2023) database in previous studies.

### 2.5. Molecular Docking Predicts ACE Inhibitory Activity of S. pilchardus Peptides

Molecular docking techniques were used to further screen the ACE inhibitory peptides. Angiotensin II can be inactivated by ACE inhibitors by competitively and specifically blocking the ACE active site, which indicates that the peptides have ACE inhibitory activity if they can bind to ACE firmly [28]. Human ACE 3D crystal structure (PDB ID: 1O86) was acquired from Protein Data Bank (https://www.rcsb.org/) (accessed on 28 March 2023). Discovery Studio 2019 was used to run molecular docking simulations. A 1O86 crystal structure was used as the acceptor. The 2D structure of the peptide was created using ChemDraw 19.0, and the 3D structure was constructed using Chem3D with energy minimization. The ligand (lisinopril) and water were removed from the 3D crystal structure of ACE using Discovery Studio 2019, and hydrogen was added. The docking site was selected to dock with the active site of lisinopril AC5, with the docking coordinates: x = 40.633957, y = 34.479313, and z = 44.730348, and the parameters were set to radius = 13 Å; visible = yes; visible locked = no; transparent = yes; other default. The Top Hits parameter was expanded, the Pose Cluster Radius was set to 0.1, the random conformations were set to 10, the orientations were set to 10, the simulated annealing was set to true, and the rest of the parameters were default. Combining the -CDOCKER ENERGY and -CDOCKER INTERACTION ENERGY values, the lower peptides were selected for the in vitro ACE inhibition activity assay.

### 2.6. In Vitro ACE Inhibitory Activity Assay

The ACE inhibitory activity assay was determined according to a previous report with some modifications [29]. HHL was dissolved in BBS to make a solution with a concentration of 5 mmol/L. The ACE was dissolved in BBS at a concentration of 0.06 U/mL. The reaction system was as follows: 20 μL of 0.06 U/mL ACE solution and 40 μL of sample solution or control (BBS) was added. After 5 min in a water bath at 37 °C, 50 μL of 5 mM HHL was added and incubated at 37 °C for 1 h. At the end of the incubation, 50 μL of 1 mmol/L HCL was added to terminate the reaction. 500 μL water was added to the reaction system. Hippuric acid (HA) was quantified after being released from the system using HPLC and InertSustain C18 (5 μm, 4.6 × 250 mm). HPLC analysis conditions were as follows: the UV wavelength was constant at 228 nm, the mobile phase was 0.1% TFA/25% ACN in water, the flow rate was 1 mL/min, and the injection volume was 10 μL. The ACE inhibition activity was calculated by the following equation:(1)$$\mathrm{ACE}\;\mathrm{Inhibition}\;\mathrm{Activity}=\frac{∆A-∆B}{∆A}\;\times \;100,$$
where ΔA represented the peak area of HA in the control group; ΔB represented the peak area of the sample group.

### 2.7. Synthetic Peptides

Synthetic peptides (25 mg, purity > 95%), quality assured by RP-HPLC, were selected for synthesis in Sangon Biotech Co., Ltd. (Shanghai, China). The synthetic peptides were stored as dry powder at −20 °C. The synthetic peptides were dissolved in deionized water when used and kept the solution at −80 °C.

### 2.8. Measurement of the ACE Inhibition Pattern of the Screened Peptides

According to the method described by Li et al. [30], the ACE inhibition pattern of the screened peptides was investigated. First, 20 μL peptides with concentrations of 0, 0.5, and 1 mg/mL were added to 20 μL of 0.06 U/mL ACE solution and incubated in a water bath at 37 °C for 5 min. Then, 50 μL of HHL solution with different concentrations (1.25, 2.5, 3.75, 5 mmol/L) was added to the reaction system. The mixture was incubated at 37 °C for 50 min and then heated at 90 °C for 10 min to end the reaction. The graph was drawn with 1/HHL as the X-value and 1/V as the Y-value. The ACE inhibition pattern was investigated based on the Lineweaver-Burk curve intercept.

### 2.9. Determination of Zinc-Chelating Capacity

Protein hydrolysates and fractions were evaluated for zinc-chelating capacity as described in a previous work [31]. The assay principle was based on the reaction of 4-(2-pyridylazo)resorcinol with zinc ions to form a red coordination complex. Solutions of the samples (1 mg/mL each peptide) and 2 mM of 4-(2-pyridylazo)resorcinol reagent were prepared in deionized water. Then, 250 μL of each peptide was mixed with 130 μL of ZnSO_4_.7H_2_O (25 mM) and 2 μL of 0.5 M DTT. For the blank, the samples were substituted with an equal volume of deionized water. The mixtures were incubated at 37 °C for 10 min. Thereafter, 20 μL of 4-(2-pyridylazo) resorcinol (2 mM) was added to the reaction mixtures to bind zinc ion, followed by absorbance measurement at 500 nm. Zinc chelating capacity was calculated as follows:(2)$$\mathrm{Zinc}\;\mathrm{Chelating}\;\mathrm{Capacity}=\frac{∆A-∆B}{∆A}\;\times \;100,$$
where ΔA is absorbance of blank; ΔB is absorbance of sample.

### 2.10. Stability of Synthetic Peptides

#### 2.10.1. HPLC for Synthetic Peptides

Peptides were quantified using HPLC and InertSustain C18 (5 μm, 4.6 × 250 mm). For FIGR, HPLC analysis conditions were as follows: the UV wavelength was constant at 228 nm, the mobile phase was 0.1% TFA/15% ACN in water, the flow rate was 1 mL/min, and the injection volume was 10 μL. For rest peptides, the mobile phase was 0.1% TFA/25% ACN in water, and the other conditions were the same as FIGR.

The peptide solutions of 0.10, 0.25, 0.50, 1.00, and 1.50 mg/mL were prepared precisely, and the peak areas of the peptides at different concentration conditions were determined. The peak area was plotted as the vertical coordinate and the concentration as the horizontal coordinate to find out the regression equation.

#### 2.10.2. Measurement of Digestive Stability

In total, 50 μL of the 10 mg/mL peptide solution was mixed with 450 μL of artificial gastric fluid or artificial intestinal fluid in a water bath at 37 °C for 2.0 h. Then, digestion was stopped in a water bath at 90 °C for 5 min, and the method for determining the peptide content was mentioned in Section 2.10.1. Peptide preservation rate was calculated as
(3)$$\mathrm{Peptide}\;\mathrm{Preservation}\;\mathrm{Rate}=\frac{\mathrm{Peptide}\;\mathrm{content}\;\mathrm{after}\;\mathrm{treatment}}{\mathrm{Peptide}\;\mathrm{content}\;\mathrm{before}\;\mathrm{treatment}}\;\times \;100,$$

### 2.11. Antioxidant Activity

#### 2.11.1. DPPH Radical Scavenging Activity

The 2,2-diphenyl-1-picrylhydrazyl (DPPH) is a stable free radical with a distinct purple color and absorbs light at 517 nm. Antioxidants added to the DPPH solution lead to a reduction in the DPPH radical to a non-radical form, resulting in a decrease in the solution’s absorbance. The greater the reduction in absorbance, the more potent the scavenging activity. This principle of DPPH radical scavenging is commonly utilized for evaluating the antioxidant capacity of various substances, such as peptides.

Sample (1 mg/mL) group was aliquots of samples being mixed 1:1 (*v*/*v*) with 0.1 mmol/L DPPH (in ethanol absolute); control group was aliquots of samples being mixed 1:1 (*v*/*v*) with ethanol absolute; and blank group was aliquots of ethanol absolute being mixed 1:1 (*v*/*v*) with 0.1 mmol/L DPPH (in ethanol absolute). The mixture was shaken and kept for 40 min at room temperature and protected from light. The radical scavenging activity of DPPH radical was determined by measuring the absorbance at 517 nm. Glutathione (GSH) at 10 μg/mL was used as the positive control for antioxidant activity. In this study, the sample concentration was 1 mg/mL, which was determined to be within the linear range of the concentration of the sample and its DPPH scavenging ability. The ability of the sample to scavenge DPPH radical was calculated according to the following equation:(4)$$\mathrm{DPPH}\;\mathrm{Radical}\;\mathrm{Scavenging}\;\mathrm{Activity}=(1\;-\;\frac{A_{i}-A_{j}}{A_{0}})\;\times \;100,$$
where *A_i_* is absorbance of sample group, *A_j_* is absorbance of control group, and *A*_0_ is absorbance of blank group.

#### 2.11.2. ABTS+ Scavenging Activity

The 2,2′-Azinobis(3-ethylbenzothiazoline-6-sulfonic acid Ammonium Salt (ABTS) is commonly utilized as a stable free radical source in experiments. It is oxidized by potassium persulfate to produce an ABTS+ radical cation, which has a blue-green hue and absorbs light at 734 nm. When antioxidants are added, the free radical cations are reduced to the colorless ABTS form.

The ABTS+ was generated by mixing 7 mM ABTS and 4.09 mM potassium persulfate in the same volume, and the mixture was kept in the dark at room temperature for 12 h before use. The sample (1 mg/mL) group added sample 0.5 mL with 1 mL of ABTS+ solution. Control group was the same as the sample group, except that the sample was replaced with deionized water. Blank group was added deionized water 0.5 mL with 1 mL of ABTS+ solution. After incubation at room temperature for 6 min, the absorbance was tested at 734 nm. The positive control was 10 μg/mL glutathione (GSH). The sample’s concentration of 1 mg/mL falls within the linear range of the sample’s ABTS scavenging ability.
(5)$$\;\mathrm{ABTS}+\mathrm{Scavenging}\;\mathrm{Activity}=(1\;-\;\frac{A_{1}-A_{2}}{A_{3}})\;\times \;100,$$
where *A*_1_ is absorbance of sample group, *A*_2_ is absorbance of control group, and *A*_3_ is absorbance of blank group.

#### 2.11.3. Hydroxyl Radical Scavenging Activity

The hydroxyl radical (·OH) is a free radical that comprises an oxygen atom and a hydrogen atom (·OH). The principle behind measuring hydroxyl radical scavenging activity involves the reaction between the hydroxyl radical and salicylic acid, which produces 2, 3-dihydroxybenzoic acid that can be quantified by measuring the absorbance at 510 nm.

The 1 mL of sample (1 mg/mL) was mixed with 1 mL of 4.5 mmol/L of ferrous sulfate and 1 mL of 30% H_2_O_2_ (*v*/*v*). The reaction mixture was standing for 10 min after vortexing. Then, 1 mL of 4.5 mmol/L salicylic acid was added and subsequently vortexed. After a water bath at 37 °C for 30 min, the absorbance of the mixture was measured at 510 nm (*A_s_*). For the control, deionized water was added instead of salicylic acid (*A_c_*). Vitamin C (Vc) was used as a positive control at 10 μg/mL. Hydroxyl radical scavenging activity was calculated as follows:(6)$$\mathrm{Hydroxyl}\;\mathrm{Radical}\;\mathrm{Scavenging}\;\mathrm{Activity}=(1\;-\;\frac{A_{c}-A_{s}}{A_{c}})\;\times \;100,$$

#### 2.11.4. Fe^2+^-Chelating Ability

When Fe^2+^ and ferrozine are combined, they form a stable magenta-colored complex. The presence of chelators in the solution can cause them to bind with the Fe^2+^ ions, thereby preventing the formation of the Fe^2+^-ferrozine complex. As a result, the amount of Fe^2+^-ferrozine complex in the solution decreases, leading to a decrease in the absorbance of the solution at 562 nm.

The sample (1 mg/mL) group was a 1 mL sample mixed with 3.7 mL deionized water and 0.1 mL of FeCl_2_ (2 mmol/L) and 0.2 mL of ferrozine (5 mmol/L). The mixture was protected from light and kept at room temperature for 10 min prior to measuring the absorbance at 562 nm. Control group contained everything, except it used deionized water instead of sample, and blank group used deionized water instead of FeCl_2_ (2 mmol/L). EDTA-2Na at a concentration of 10 μg/mL was used as the positive control to compare the chelating ability of the samples. The chelating ability was calculated according to the following equation:(7)$${\mathrm{Fe}}^{2+}\text{-}\mathrm{chelating}\;\mathrm{ability}=(1\;-\;\frac{A_{1}-A_{2}}{A_{3}})\;\times \;100,$$
where *A*_1_ is absorbance of sample group, *A*_2_ is absorbance of blank group, and *A*_3_ is absorbance of control group.

### 2.12. Statistical Analysis

All experiments were performed in triplicate (except for the molecular docking results), and data were reported as mean ± SD (n ≥ 3). An ANOVA test using SPSS 26.0 (SPSS Corporation, Chicago, IL, USA) was used to analyze the experimental data. The IC_50_ value was defined as the half concentration of the peptide required to inhibit the ACE activity, and the IC_50_ of the ACE inhibitory peptides was determined by calculating the ACE inhibition rate at various sample concentrations using the SPSS 26.0 probit regression model.

## 3. Results and Discussion

### 3.1. Biological Activities of Bioactive Peptides from SPH and Ultrafiltered Fractions

#### 3.1.1. ACE Inhibitory Activity of SPH and Ultrafiltration Fraction

ACE inhibitory peptides have been shown to play a very crucial role in regulating blood pressure. In this study, we observed a significant increase in ACE inhibitory activity (*p* < 0.05) from high-molecular-weight (MW) to low-MW peptides, as shown in Figure 1. The IC_50_ of SPH was 96.09 ± 9.59 μg/mL. The IC_50_ of the <5 kDa fraction was 79.59 ± 3.40 μg/mL, and the IC_50_ of the >5 kDa fraction was 200.96 ± 1.38 μg/mL, which was much higher than the IC_50_ of the <5 kDa fraction. IC_50_ of the <3 kDa fraction was 67.88 ± 1.50 μg/mL, and IC_50_ of the 3–5 kDa fraction was 112.25 ± 4.18 μg/mL, which was significantly higher than IC_50_ of the <3 kDa fraction. The results are consistent with previous findings that suggest low-MW peptides (<3 kDa) seem to be more effective for inhibiting ACE activity than high-MW peptides [32]. These low molecular weight ACE-inhibitory peptides have a high potential for lowering blood pressure. These findings are important for the design and manufacture of foods with antihypertensive functions and can provide new ideas and methods for the development of functional foods.

It has been reported that *S. pilchardus* are a good source of ACE inhibitors. Matsufuji et al. used alkaline protease from *Bacillus licheniformis*, which was obtained from Novo Co. (2.4 L, type FG), to hydrolyze *S. pilchardus* muscle and obtained an IC_50_ of 0.26 mg/mL. [18]; Jemil et al. hydrolyzed *S. pilchardus aurita* muscle with *Bacillus subtilis* A26 proteases, and the IC_50_ was 0.16 mg/mL [20]; Pedro et al. hydrolyzed whole *S. pilchardus* by subtilisin for 2 h and trypsin for 2 h, and the IC_50_ was 439 ± 16 μg/mL; when hydrolyzed by trypsin for 2 h and subtilisin for 2 h, the IC_50_ was 442 ± 25 μg/mL; when hydrolyzed by Subtilisin and Trypsin for 4 h, the IC_50_ was 489 ± 22 μg/mL [22]; Bougatef et al. hydrolyzed *S. pilchardus aurita* head and viscera by Proteases NH_1_, and the IC_50_ was 2.1 ± 0.36 mg/mL; when hydrolyzed by alcalase, the IC_50_ was 2.3 ± 0.24 mg/mL; when hydrolyzed by protease from *S. pilchardus*, the IC_50_ was 1.2 ± 0.09 mg/mL; when hydrolyzed by chymotrypsin, the IC_50_ was 1.8 ± 0.13 mg/mL; when hydrolyzed by proteases ES_1_, the IC_50_ was 7.4 ± 0.62 mg/mL [24]; Oscar et al. hydrolyzed heads and visceras of *S. pilchardus* by alkaline protease, and the IC_50_ was 1.16 ± 0. 04 mg/mL [25]; Vieira et al. made muscles from *S. pilchardus* hydrolyzed by the brewer’s spent yeast (Bsy) proteases, and the IC_50_ was 984 μg/mL; when hydrolyzed by the neutrase, the IC_50_, was 735 μg/mL; when hydrolyzed by the alcalase, the IC_50_ was 619 μg/mL [26]. In this study, dispase and alkaline protease were used to hydrolyze *S. pilchardus*, resulting in ACE inhibitory peptides with IC_50_ values superior to those reported in previous studies. Therefore, this research has contributed to the advancement of methods for producing ACE inhibitory peptides from *S. pilchardus*.

The IC_50_ values for the ACE inhibitory activity of the <3 kDa peptide fraction from SPH in our study were found to be higher than those observed for the protein hydrolysate from *Agaricus bisporus scraps* (0.9 mg/mL) collected from Tianshui Zhong Xing Fungi Technology Co., Ltd. (Tianshui, China) [33] and ginkgo protein (0.224 mg/mL) [12]. This result suggests that low-MW peptide < 3 kDa from ultrafiltration separation exhibits high ACE inhibitory activity and could be used as a natural ACE inhibition agent, which was considered to be a useful therapeutic approach to treat hypertension. Accordingly, we further screened the <3 kDa fraction of SPH based on its physicochemical properties, toxicity prediction, and molecular docking predictions.

#### 3.1.2. Identification (LC-MS/MS) and Screening (Online Databases and Molecular Docking) Were Performed to Identify ACE Inhibitory Peptides from *S. pilchardus*

In this study, a combination of LC-MS/MS and bioinformatics analysis could simultaneously identify all peptide components of hydrolysates. A total of 1449 sequences were identified from the <3 kDa fraction of SPH; the number and peak area distribution of peptides obtained are shown in Figure 2A. As shown in Table 1, the potential biological activity of each peptide was evaluated, and peptides with scores above 0.8, non-toxic, and good water solubility were selected for further evaluation. The results showed that oral administration of this peptide had little possibility of safety concerns and was suitable for subsequent processing.

The results of Table 1 showed that 37 of the 1449 sequences detected by LC-MS/MS meet the standard. In total, 16.22% of them were tripeptides, 81.08% were tetrapeptides, and 1% were pentapeptides. This was consistent with the report of Suo et al. that peptides with shorter sequences and smaller molecular masses had stronger biological activity [34]. The relationship between peptide sequences and the molecular mass is shown in Figure 2B. The results showed that tripeptides and tetrapeptides were more likely to be physiologically active, as evidenced by the results of this study.

The peptides shown in Table 1 were further screened by molecular docking; all 11 eligible peptides had a higher -CDOCKER ENERGY than Lisinopril. As shown in Table 2, they were LDGF, FRAL, FQRL, LDPF, FMPK, FILR, FIGR, FDRL, DLMF, KLF, and KFL. Among the above peptides, all the other 10 peptides except KFL had a higher -CDOCKER INTERACTION ENERGY than that of Lisinopril. Searching the AHTPDB database, none of the 11 sequences were recorded as ACE inhibitory peptides. -CDOCKER ENERGY and -CDOCKER INTERACTION ENERGY were scoring functions for the CDOCKER algorithm in Discover Studio 2019 Client.

-CDOCKER ENERGY represented the ligand strain energy and receptor-ligand interaction energy, and -CDOCKER INTERACTION ENERGY was a non-bonded interaction that existed between the protein and the ligand [14]. With both scores taken into account, 11 peptides from Table 2 were chosen for synthesis and verification: LDGF, FRAL, FQRL, LDPF, FMPK, FILR, FIGR, FDRL, DLMF, KLF, and KFL.

#### 3.1.3. ACE Inhibition Activities and Zinc Chelating Capacity of Synthetic Peptide

Our findings demonstrated that the sequences FIGR, FILR, FQRL, FRAL, KFL, and KLF have substantial ACE inhibitory action. The ACE-inhibitory capacity was highly dependent on the peptide sequence. The IC_50_ values of the above sequence are shown in Figure 3A.

Zinc plays an essential role in the catalytic activity of ACE, and molecules that chelate Zn^2+^ ions can partially reduce its activity, and in some cases, complete inhibition can yield the inactive. The zinc chelating capacity of synthetic peptides is shown in Figure 3B.

The results showed that the chelating ability of the peptides varied, with FQRL showing the highest chelating ability at 93.81% ± 2.91, followed by FILR at 86.98% ± 2.29. The peptides FIGR, FRAL, KFL, and KLF showed lower chelating abilities, with values ranging from 73.97% ± 2.18 to 75.40% ± 3.43.

Among the six peptides tested, FQRL showed the highest chelating ability, which could be attributed to the presence of glutamine residues in its sequence. Glutamine has been shown to have a high affinity with Zn^2+^ [35]. On the other hand, the peptides FIGR, FRAL, KFL, and KLF showed lower chelating abilities, which could be due to the absence or lower abundance of metal-binding amino acid residues in their sequences; this is consistent with molecular docking results. Overall, all six peptides showed excellent zinc chelating capacity, with FILR and FQRL having the highest zinc chelating capacity, while these two peptides also possessed the best ACE inhibition ability, which was consistent with the trend of zinc chelating capacity.

The results of the study show that the IC_50_ values for ACE inhibition of six synthetic peptides, namely FIGR, FILR, FQRL, FRAL, KFL, and KLF, were determined to be 0.60 ± 0.01 mg/mL, 1.01 ± 0.02 mg/mL, 1.54 ± 0.14 mg/mL, 0.66 ± 0.09 mg/mL, 0.61 ± 0.02 mg/mL, and 0.89 ± 0.01 mg/mL, respectively. These findings suggest that the peptides FIGR, FRAL, and KLF are more potent ACE inhibitors, while FQRL has the least potency. The ACE inhibitory activity of KLF was similar to that of FILR, probably due to the smaller molecular mass [36]. However, KFL and KLF have the same molecular mass, and the difference in ACE inhibitory activity may be related to the C-terminal amino acid. According to published works, ACE inhibitory peptides preferably had a hydrophobic amino acid at their C-terminus, such as tryptophan (W), tyrosine (Y), and phenylalanine (F) [37].

In the previous study, Hu et al. showed that the IC_50_ of GLLGY, HWP, and VYGF for ACE inhibition was 1 mg/mL, and the IC_50_ of HWP for pancreatic lipase was 3.95 mg/mL [38]. Chen et al. suggested that AEYLCEAC (IC_50_ = 4.287 mM) might act as a helpful ingredient in functional foods or pharmaceuticals for the prevention and treatment of hypertension [39]. These results revealed that the peptides FIGR, FILR, FQRL, FRAL, KFL, and KLF might be suitable candidates for further research on ACE inhibition.

The study provides valuable insights into the ACE inhibition activity of synthetic peptides and suggests potential avenues for the development of new ACE inhibitors. This study provides new ideas for optimizing the extraction and purification methods of ACE-inhibitory peptides, which can provide an important reference value for future research. Further research is warranted to validate these findings and explore the therapeutic potential of these peptides. However, it is important to note that these results were obtained in vitro, and further studies are needed to determine their efficacy and safety in vivo. Additionally, the study only examined the ACE inhibition activity of these peptides and did not investigate their potential side effects or interactions with other drugs. Therefore, caution should be exercised before considering the use of these peptides as therapeutic agents.

#### 3.1.4. Inhibition Mechanism of Peptides in Molecular Docking

Two hydrogen bonds at different distances were formed between FIGR and the amino acid residues ALA356 and ALA354 of ACE. It was discovered that HIS383, VAL380, PHE457, and TYR523 were the main residue that interacted between FIGR and ACE, and π-Alkyl interaction was crucial to the binding affinity between the peptide and ACE. The π-Cation interaction between FIGR and HIS387 promoted binding between the peptide and ACE. The Attractive Charge interaction between FIGR and ARG522, GLU411, and ASP453 also facilitated the binding between this peptide and ACE. FIGR also was involved in the Salt Bridge interaction with GLU376. TYR523, ALA354, GLU384 belong to pocket S1. The oxygen atoms in the FIGR and Zn^2+^ participated in Attractive Charge interactions (Figure 4A).

Three hydrogen bonds were formed between FILR and the amino acid residues GLN281, ALA356, and ALA354 of ACE. FILR also involved in the formation of π-Alkyl interactions with PHE512, TYR523, HIS383, HIS513, and HIS353; Alkyl interactions with VAL518; π-Cation interactions with ARG522, HIS387, and HIS410; Attractive Charge with GLU411, GLU384, and GLU376 Attractive Charge; Salt Bridge with GLU162, ASP377, and LYS511; and pi-pi T-shape with HIS410. GLU384, TYR523, and ALA354 belong to pocket S1; GLN281, LYS511, HIS513, and HIS353 belong to pocket S2; and GLU162 belongs to pocket S3. The oxygen atoms in the FILR and Zn^2+^ participated in Metal-Acceptor interactions (Figure 4B).

Four hydrogen bonds were formed between FQRL and the amino acid residues ALA354, TYR520, ARG522, and LYS511 of ACE. FQRL was involved in π-Alkyl interaction with PHE527, PHE457, TYR523, and π-Cation interaction with ARG522. Attractive Charge interaction with ASP377, GLU376, GLU162, and LYS511; Salt Bridge interaction with GLU162. The TYR523 and ALA354 belong to pocket S1, the TYR520 and LYS511 to pocket S2, and the GLU162 to pocket S3. The oxygen atoms in the FQRL and Zn^2+^ participated in Metal-Acceptor interactions (Figure 4C).

Four hydrogen bonds were formed between FRAL and the amino acid residues ALA354, GLN281, TYR520, and GLU384 of ACE. FRAL was involved in the formation of π-Alkyl with PHE512 and VAL379; forms Alkyl with VAL518; forms π-Cation with PHE457; forms Attractive Charge with GLU376, ASP415, ASP377, and GLU162; and forms Salt Bridge with ASP377 and GLU162. GLU384 and ALA354 belong to pocket S1, GLN281 and TYR520 to pocket S2, and GLU162 to pocket S3. The oxygen atoms in the FRAL and Zn^2+^ participated in Attractive Charge interactions (Figure 4D).

Three hydrogen bonds were formed between KFL and the amino acid residues ALA354, CYS370, and GLU384 of ACE; KFL forms Alkyl interaction with VAL518; and forms an Attractive Charge with ASP377, GLU376, and GLU162. GLU384 and ALA354 belong to pocket S1 and GLU162 to pocket S3. The oxygen atoms in the KFL and Zn^2+^ participated in Attractive Charge interactions (Figure 4E).

Three hydrogen bonds are formed between KLF and amino acid residues CYS370, THR372, ALA354, and GLU384 of ACE. KLF is involved in π-Alkyl interaction with VAL518 and HIS383; forms Alkyl with VAL380; forms π-Cation with HIS353; and forms Attractive Charge and Salt Bridge with GLU162 and ASP377. ALA354 and GLU384 belong to pocket S1, HIS353 to pocket S2, and GLU162 to pocket S3. The oxygen atoms in the KLF and Zn^2+^ participated in Attractive Charge interactions (Figure 4F).

In contrast to the Attractive Charge of Zn^2+^ in ACE with FIGR, FRAL, KFL, and KLF, the Zn^2+^ in ACE was linked by Metal-Acceptor with FILR and FQRL, consistent with the content of Section 3.1.3.

These interactions changed the ACE structure, which inhibited ACE activity, confirming previous research [40,41].

#### 3.1.5. Inhibition Pattern of Peptides

Using four substrate concentrations (1.25, 2.5, 3.75, and 5 mmol/L HHL) and three inhibitory peptide concentrations (0, 0.5, and 1 mg/mL), the inhibition pattern of six ACE inhibitory peptides were determined by Lineweaver-Burk plots to further clarify the mechanism of interaction of these peptides on ACE. According to Figure 5, the results for all sequences showed three straight lines intersected at one point on the y-axis, indicating that all six ACE inhibitory peptides were competitive inhibitors. This result showed that all six peptides were bound competitively to critical residues in the ACE catalytic region to form ACE-peptide complexes, thus reducing the affinity of ACE for HHL. This result was consistent with the molecular docking results, which demonstrated that FIGR was associated with the S1 pocket, the KFL bond was associated with the S1 and S1’ pockets, and the remaining peptides were associated with all three pockets, as explained in Section 3.1.4.

Competitive ACE inhibitory peptides have been identified in many reports, including peptide sequence WPMGF from shellfish [42], VAP isolated from grass carp [43], FHAPWK extracted from black cassia seeds [44], RWDISQPY from Sargassum [45], and YQK isolated from bovine casein [46]. In addition, all six inhibitory peptides contained hydrophobic amino acids, of which phenylalanine and leucine occupied most of the N-terminus or C-terminus. Additionally, competitive inhibition of the peptides has been reported to be associated with the presence of hydrophobic amino acids at the N-terminus or C-terminus of the peptide [47]. This report may explain the competitive inhibition of all six ACE inhibitory peptides.

#### 3.1.6. Antioxidant Activities of Synthetic Peptide, SPH, and Ultrafiltration Fraction

The structure of six peptides was studied, and it was found that all six peptides contain phenylalanine. Previous research has shown that peptide segments containing phenylalanine are more likely to possess antioxidant activity [7,48], which may be related to the aromatic ring [49]. Therefore, it is necessary to determine the antioxidant activity of these six peptides. In this study, we evaluated the antioxidant capacities of synthetic peptides using three different assays: ABTS, DPPH, and Fe^2+^ chelating assays. The results are summarized in Figure 6. Antioxidant activity was measured at the same peptide concentration (1 mg/mL), all peptides had antioxidant activity, and the positive control was measured at 10 μg/mL.

For the ABTS assay, the percentage of ABTS clearance was measured for six different peptides: FIGR, FILR, FQRL, FRAL, KFL, and KLF. The results showed that the peptides had varying degrees of ABTS radical scavenging capacity, with FILR exhibiting the highest clearance percentage of 87.53% ± 3.31, followed by FIGR at 82.53% ± 3.90. The other four peptides showed lower clearance percentages ranging from 64.35% ± 5.38 to 65.70% ± 6.21. In comparison, the positive control 10 μg/mL GSH showed a much lower ABTS scavenging capacity of 9.01% ± 0.69.

For the DPPH assay, we also measured the percentage of DPPH clearance capacity for the six peptides. The results showed that all six peptides had DPPH radical scavenging capacity, with FQRL exhibiting the highest clearance percentage of 42.53% ± 1.80, followed by KLF at 35.62% ± 0.14. The other four peptides showed lower clearance percentages ranging from 23.05% ± 0.37 to 38.53% ± 1.10. In comparison, the positive control 10 μg/mL GSH had a DPPH scavenging capacity of 33.25% ± 1.44.

For the Fe^2+^ chelating assay, we measured the percentage of Fe^2+^ chelating capacity for the six peptides. The results showed that the peptides had varying degrees of Fe^2+^ chelating capacity, with FRAL exhibiting the highest chelating percentage of 15.65% ± 0.44, followed by FIGR at 15.49% ± 0.59. The other four peptides showed lower chelating percentages ranging from 5.71% ± 0.66 to 14.47% ± 0.45. In comparison, the positive control 10 μg/mL EDTA-2Na had a much higher Fe^2+^ chelating capacity of 28.81% ± 1.13.

The hydroxyl radical scavenging ability of synthetic peptides was also measured, but the synthetic peptide did not exhibit hydroxyl radical scavenging ability.

Overall, the results suggest that the synthetic peptides have varying degrees of antioxidant capacity, and their efficacy is dependent on the type of radical and assay used. This paper provides new ideas and approaches for the development of antioxidant peptides and ACE inhibitors with pharmacological applications. These findings may have important implications for the development of novel therapeutic agents with antioxidant properties.

The antioxidant activity of SPH and ultrafiltration fraction was determined. The antioxidant capacities were determined by ABTS, DPPH, hydroxyl radical scavenging ability, and Fe^2+^ chelating ability. The results of the study indicated that the SPH and its ultrafiltration fraction possess potent antioxidant activities (Figure 7).

In the ABTS radical scavenging ability, the percentage for <3 kDa, 3–5 kDa, <5 kDa, >5 kDa, and SPH were 97.26% ± 0.22, 95.44% ± 0.38, 93.78% ± 0.18, 84.07% ± 0.31, and 90.74% ± 0.36, respectively. These results indicate that the <3 kDa and 3–5 kDa fractions had the highest ABTS radical scavenging ability. The positive control, 10 μg/mL GSH, had an ABTS scavenging capacity of 9.01% ± 0.69.

In the DPPH radical scavenging ability, the percentage of <3 kDa, 3–5 kDa, <5 kDa, >5 kDa, and SPH were 60.86% ± 2.24, 68.93% ± 0.40, 65.47% ± 0.29, 70.34% ± 1.13, and 51.31% ± 0.91, respectively. These results indicate that the 3–5 kDa and >5 kDa fractions had the highest DPPH radical scavenging ability. The positive control, 10 μg/mL GSH, had a DPPH scavenging capacity of 33.25% ± 1.44.

In the Fe^2+^ chelating ability, the percentage of <3 kDa, 3–5 kDa, <5 kDa, >5 kDa, and SPH were 37.44% ± 2.36, 45.73% ± 1.74, 45.91% ± 1.29, 15.84% ± 1.54, and 36.30% ± 1.78, respectively. These results indicate that the 3–5 kDa and <5 kDa fractions had the highest Fe^2+^ chelating capacity. The positive control, 10 μg/mL EDTA-2Na, had Fe^2+^ chelating capacity of 28.81% ± 1.13.

In the hydroxyl radical scavenging assay, the percentage of <3 kDa, 3–5 kDa, <5 kDa, >5 kDa, and SPH were 12.63% ± 1.03, 10.53% ± 0.44, 11.06% ± 0.65, 7.10% ± 0.40, and 15.18% ± 0.34, respectively. These results indicate that the <3 kDa and SPH fractions had the highest hydroxyl radical scavenging ability. The positive control, 10 μg/mL Vc, had a hydroxyl radical scavenging ability of 9.39% ± 0.54.

In summary, these results suggest that the different fractions of the SPH possess varying degrees of antioxidant capacities, with the 3–5 kDa and <5 kDa fractions showing the highest Fe^2+^ chelating capacity, the <3 kDa fraction showing the highest ABTS radical scavenging ability, and the >5 kDa fraction showing the highest hydroxyl radical scavenging ability. The findings of this study suggest that the SPH and ultrafiltration fraction could potentially be used as natural antioxidants in various food and pharmaceutical applications.

#### 3.1.7. Analysis of Digestive Stability by Synthetic Peptides

Gastrointestinal digestion was simulated in vitro to determine the stability of the peptides after oral administration. Some ACE-inhibiting peptides failed to show hypotensive activity after oral administration in vivo, as these peptides may be hydrolyzed by gastrointestinal digestive enzymes [50]. To evaluate the stability of the purified peptide under gastrointestinal enzyme digestion, the purified peptide was first incubated with artificial gastric fluid or artificial intestinal fluid, then subjected to HPLC profile comparisons. The results are shown in Figure 8. After gastric digestion, the retention of the peptide was close to 100%, indicating that the purified peptide was resistant to gastric digestion and that the active sequence of the peptide was not destroyed by pepsin. After intestinal digestion, the retention rate was higher except for FQRL and FRAL. The retention rates were 50.78% ± 0.77 for FIGR, 86.23% ± 1.60 for FILR, 73.36% ± 0.86 for KFL, 88.92% ± 2.60 for KLF, 25.62% ± 2.01 for FQRL, and 10.83% ± 0.44 for FRAL. Overall, the peptides remained well retained after gastric and intestinal digestion, suggesting that synthetic peptides have the potential to be incorporated into functional foods.

## 4. Conclusions

SPH demonstrated potent ACE inhibitory activity and good antioxidant activity, with the <3 kDa fraction exhibiting the most promising results. Building on this, we have established a rapid screening method for ACE inhibitory peptides, utilizing LC-MS/MS combined with online databases and molecular docking. Through this approach, we have identified six novel peptides with ACE inhibitory activity, including FIGR, FILR, FQRL, FRAL, KFL, and KLF, which also possess antioxidant capacity. All six peptides possessed zinc-chelating capacity, and competitive inhibition was also observed, and molecular docking confirmed their binding to the Zn^2+^ of ACE and to three active pockets (S1, S2, and S1’) of ACE. These findings highlight the effectiveness of our method for screening ACE inhibitory peptides and provide valuable technical support for further value-added applications of *S. pilchardus.*

## Figures and Tables

**Figure 1 foods-12-02216-f001:**
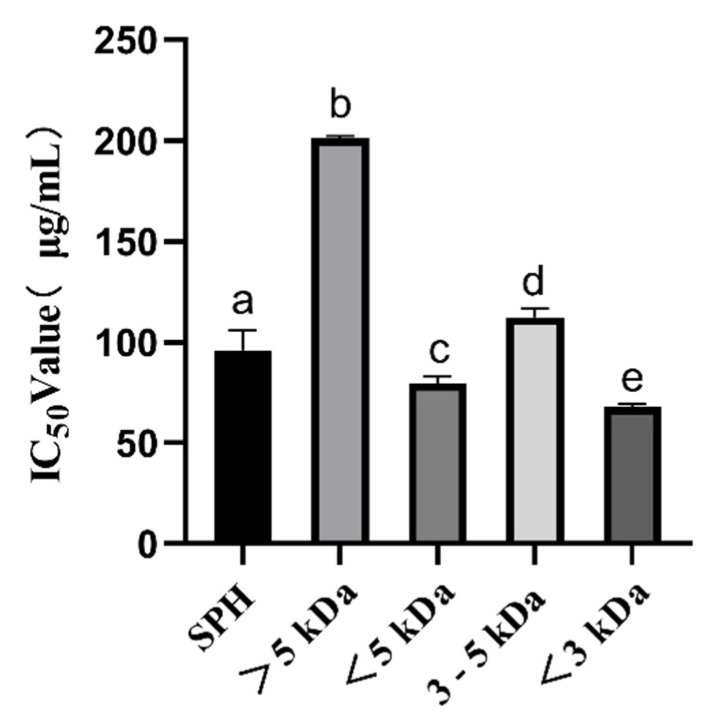
ACE inhibition IC_50_ values (μg/mL) of SPH and ultrafiltration fractions. a–e The values with the same letters indicate no significant difference (*p* > 0.05).

**Figure 2 foods-12-02216-f002:**
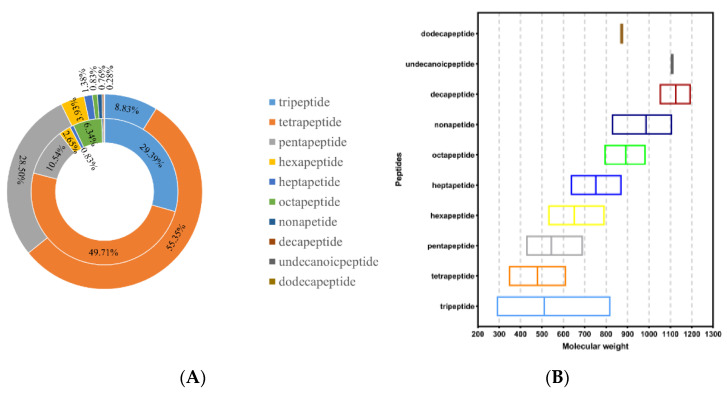
(**A**) Distribution of peptide number and peak area in <3 kDa fractions of SPH. Note: The internal pie plot reflects the MS areas, and the external pie plot reflects the numbers of peptide sequences. (**B**) Molecular mass of peptide in <3 kDa fractions of SPH (the middle line is the average value).

**Figure 3 foods-12-02216-f003:**
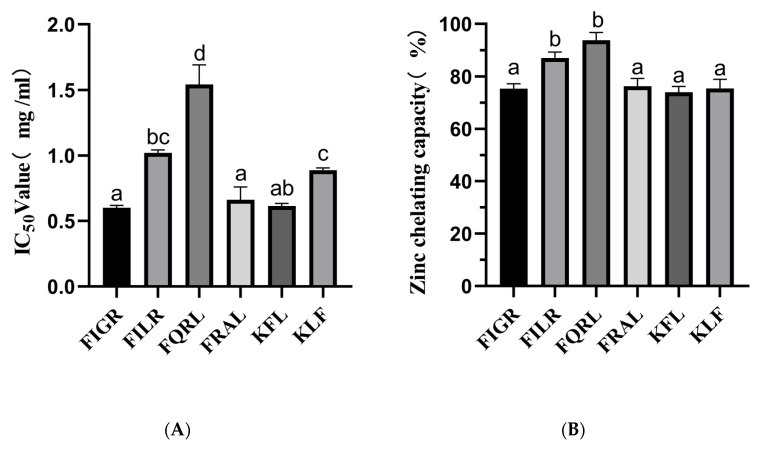
(**A**) ACE inhibitory activity of synthetic peptides; (**B**) Zinc chelating capacity of synthetic peptides. a–d The values with the same letters indicate no significant difference (*p* > 0.05).

**Figure 4 foods-12-02216-f004:**
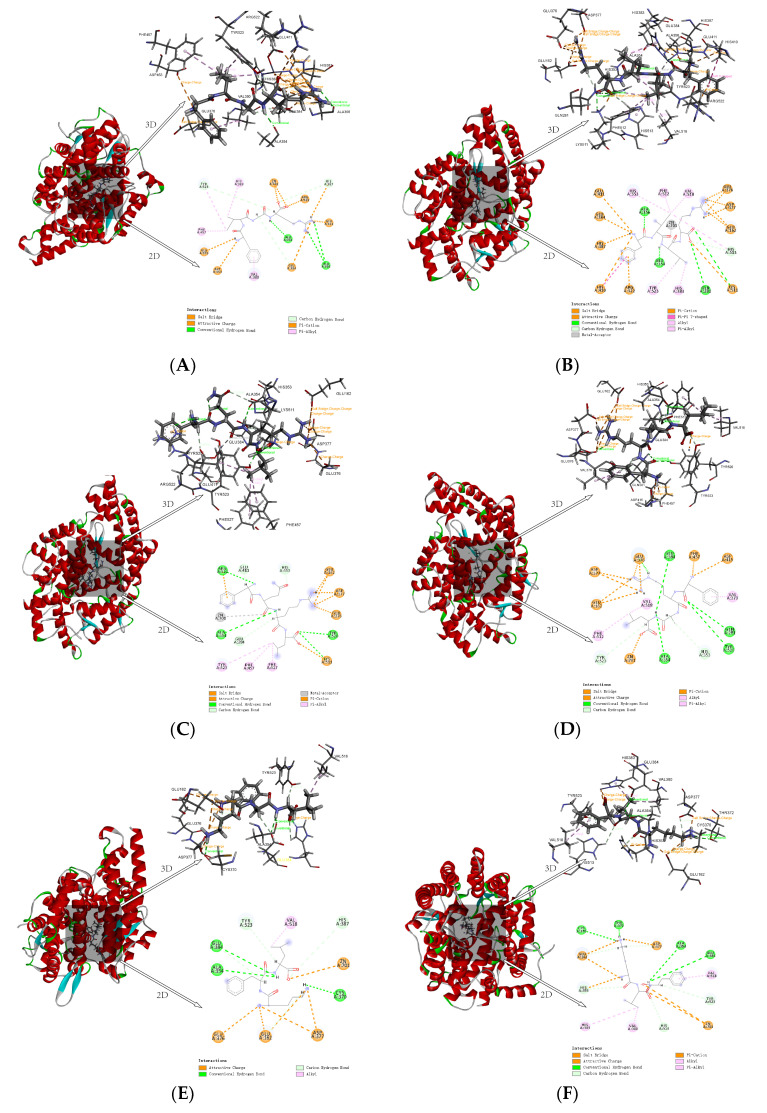
Molecular interaction between inhibitory peptides and human ACE, (**A**) FIGR, (**B**) FILR, (**C**) FQRL, (**D**) FRAL, (**E**) KFL, and (**F**) KLF.

**Figure 5 foods-12-02216-f005:**
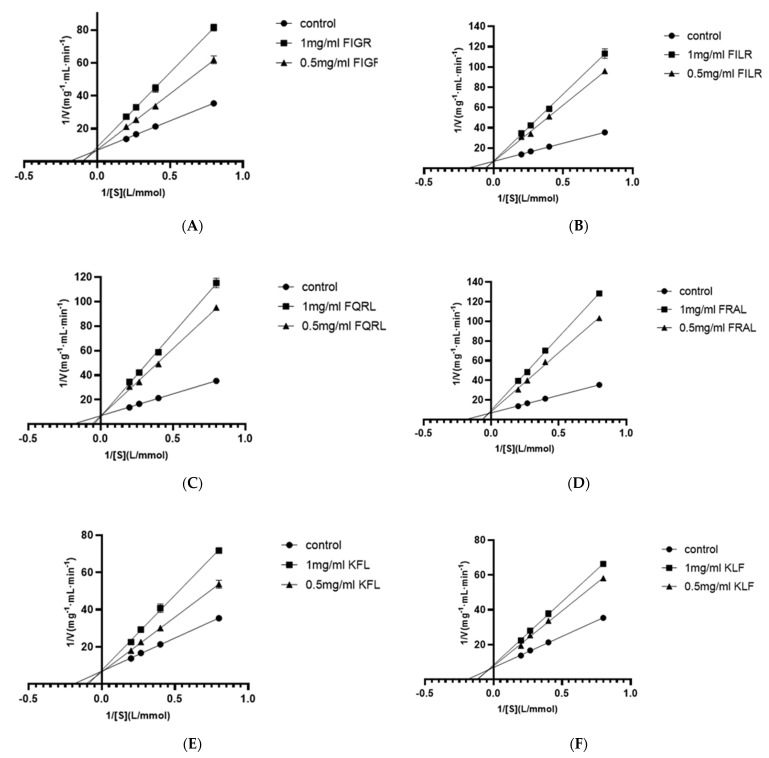
Lineweaver–Burk plot of ACE inhibitory peptides (**A**) FIGR, (**B**) FILR, (**C**) FQRL, (**D**) FRAL, (**E**) KFL, and (**F**) KLF.

**Figure 6 foods-12-02216-f006:**
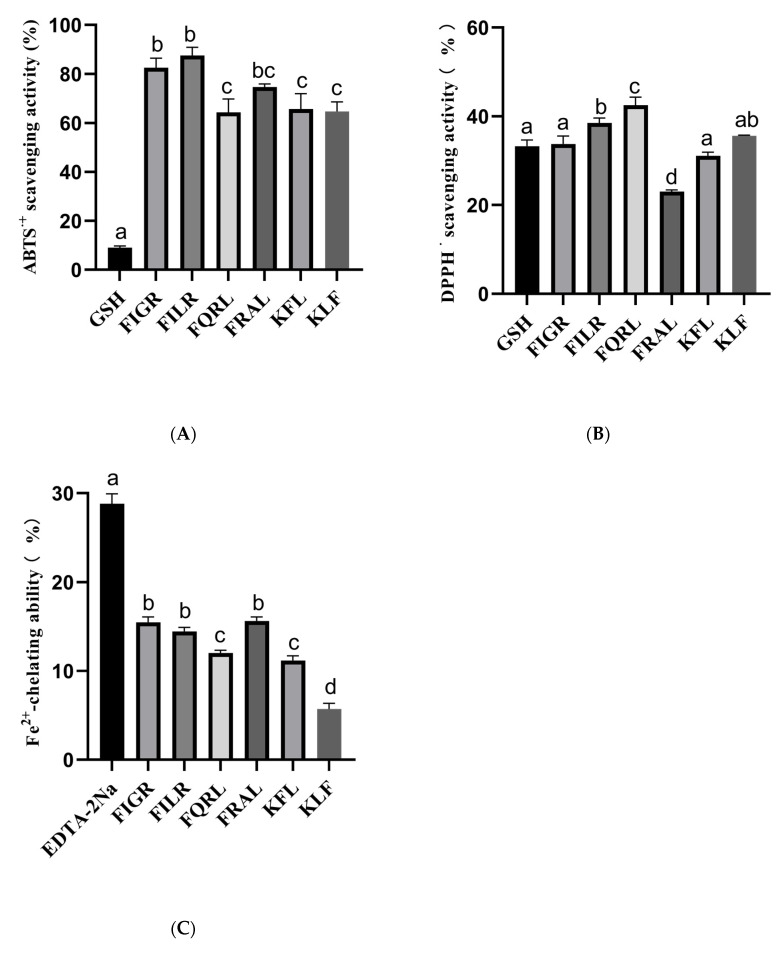
(**A**) ABTS radical scavenging activity of synthetic peptides; (**B**) DPPH radical scavenging activity of synthetic peptides; (**C**) Fe^2+^-chelating ability of synthetic peptides. a–d The values with the same letters indicate no significant difference (*p* > 0.05).

**Figure 7 foods-12-02216-f007:**
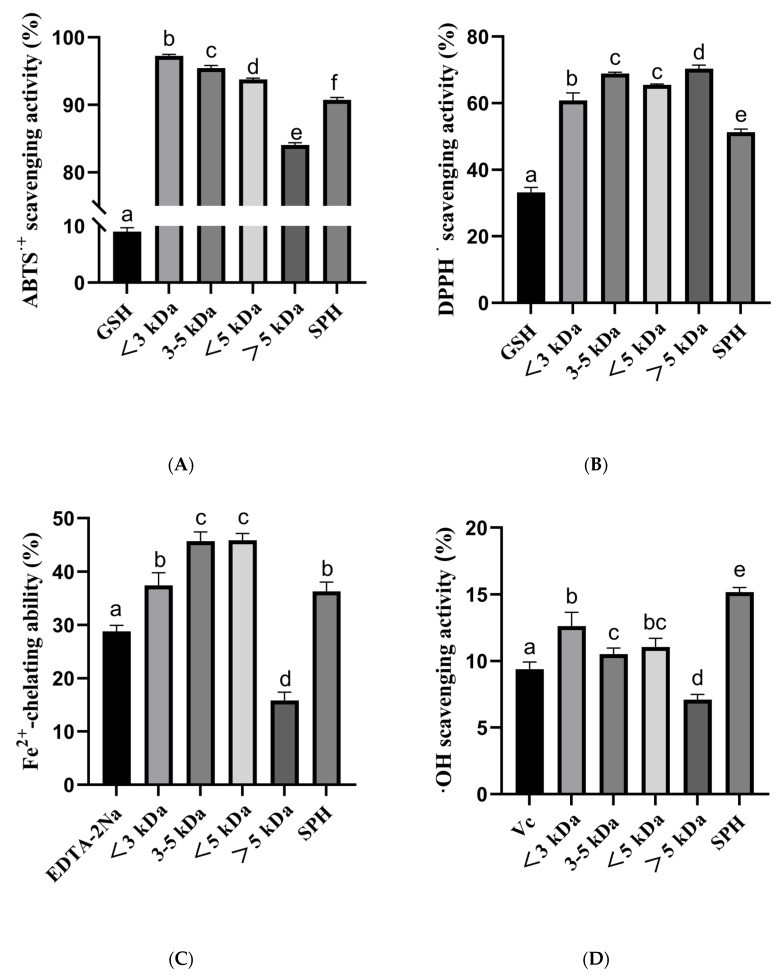
(**A**) ABTS radical scavenging activity of SHP and ultrafiltered fractions (<3 kDa, 3–5 kDa, <5 kDa, >5 kDa); (**B**) DPPH radical scavenging activity of SHP and ultrafiltered fractions; (**C**) Fe^2+^-chelating ability of SHP and ultrafiltered fractions; (**D**) Hydroxyl radical scavenging activity of SHP and ultrafiltered fractions. a–f The values with the same letters indicate no significant difference (*p* > 0.05).

**Figure 8 foods-12-02216-f008:**
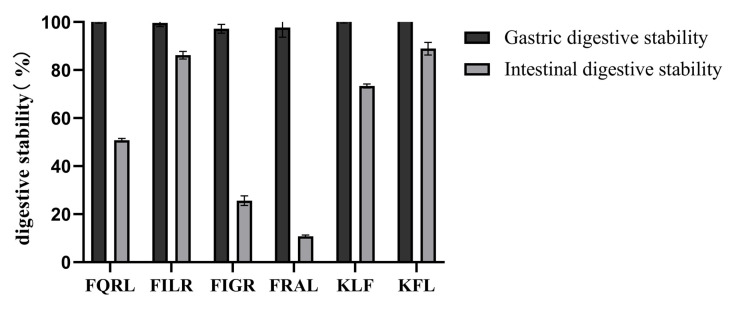
Digestive stability of synthetic peptides.

**Table 1 foods-12-02216-t001:** Potentially bioactive peptide sequences with a score above 0.8, non-toxic, and with good solubility.

Sequence	Toxicity	Solubility	Score	Sequence	Toxicity	Solubility	Score
KLF	Non-Toxic	Good	0.81	DLMF	Non-Toxic	Good	0.94
LDF	Non-Toxic	Good	0.84	FDRL	Non-Toxic	Good	0.86
FIR	Non-Toxic	Good	0.89	FIGR	Non-Toxic	Good	0.88
IFR	Non-Toxic	Good	0.90	IGRF	Non-Toxic	Good	0.93
RYF	Non-Toxic	Good	0.94	IRFL	Non-Toxic	Good	0.88
KFL	Non-Toxic	Good	0.83	LDGF	Non-Toxic	Good	0.88
FTDF	Non-Toxic	Good	0.88	SLRF	Non-Toxic	Good	0.89
LDLF	Non-Toxic	Good	0.81	WQFK	Non-Toxic	Good	0.90
DFLL	Non-Toxic	Good	0.86	LDPF	Non-Toxic	Good	0.91
FILR	Non-Toxic	Good	0.82	AFPR	Non-Toxic	Good	0.92
LFPK	Non-Toxic	Good	0.82	FDPL	Non-Toxic	Good	0.92
FEHF	Non-Toxic	Good	0.83	RLPF	Non-Toxic	Good	0.95
FQRL	Non-Toxic	Good	0.84	LRFP	Non-Toxic	Good	0.93
QYFR	Non-Toxic	Good	0.83	FMPK	Non-Toxic	Good	0.94
FGPR	Non-Toxic	Good	0.96	IFDF	Non-Toxic	Good	0.95
KDPF	Non-Toxic	Good	0.84	FLFR	Non-Toxic	Good	0.98
LDYF	Non-Toxic	Good	0.84	FDMF	Non-Toxic	Good	0.99
KYFP	Non-Toxic	Good	0.85	PWRAP	Non-Toxic	Good	0.91
FRAL	Non-Toxic	Good	0.85				

**Table 2 foods-12-02216-t002:** Eligible peptides in the molecular docking simulation.

Sequence	-CDOCKER ENERGY (kcal/mol)	-CDOCKER INTERACTION ENERGY (kcal/mol)
LDGF	121.880	107.296
FRAL	103.881	101.421
FQRL	111.194	104.606
LDPF	100.851	107.304
FMPK	105.286	109.274
FILR	93.854	103.268
FIGR	106.429	96.410
FDRL	129.310	110.089
DLMF	113.891	100.233
KLF	99.030	104.047
KFL	99.863	96.555
LPR	93.753	95.979

## Data Availability

Data is contained within the article.

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
