# Peer review of "Identification and Characterization of Novel ACE Inhibitory and Antioxidant Peptides from Sardina pilchardus Hydrolysate"

_foods, 2023, doi:10.3390/foods12112216_

Round 1

Reviewer 1 Report

This work is interesting and significant:

Lines 11-15: These sentences are not so clear. Please, rephrase them.

Line 36: Hypertension is important health concerns not only in China, but worldwide.  

Line 47: Please, put reference.

Line 113: What is dispersant? Dispase? Please correct or explain.

Line 142: Please, put respectively at the end of sentence.

Lines 188 and 208: What is the difference between sections 2.6 and 2.8. Please, clarify.

Line 230: The section 2.10.1 is related to chemically synthesized peptides?

Line 318: Only results? I think it should be Results and Discussions

Figures 1 and 2: Synthesized peptides has lower ACE inhibition activity compared to SPH. Why? Using in silico approach and mass spectrometry, the authors should identify the peptides with highest ACE inhibition potential and validate their inhibition activity in vitro.

Figures 2: Zinc chelating capacity is expressed as % with the maximum activity of around 1 %, which is very strange. I think the scale should be between 1 and 100%, not between 0 and 1%. The same applies for figures 6, 7 and 8. Please, clarify or correct it.

Figure 2: Please express IC50 values in µg/mL to be comparable to figure 1.

Figure 1: Please, provide figure with zinc chelating capacity of SPH.

Some sentences should be rephrased.

Reviewer 2 Report

This article is particularly noteworthy for its methodological rigor and its significant contribution to the field of functional food biochemistry. The authors provide valuable insights into the potential use of Sardina pilchardus as a source of bioactive peptides, specifically in relation to angiotensin-converting enzyme (ACE) inhibitory activity.

The utilization of the LC-MS/MS rapid screening strategy, combined with an online database and molecular docking, showcases an innovative and accurate approach for the discovery of novel ACE inhibitory peptides. This methodology could have far-reaching implications in the field, and the authors should be commended for their innovative approach.

Furthermore, the identification and experimental validation of 37 peptides with potential ACE inhibitory activity represents a significant and novel addition to our understanding of the bioactive potential of Sardina pilchardus. The additional finding of antioxidant activities within these peptides is of great interest and adds an extra dimension to the work.

The article is well-structured, and the findings are clearly presented. However, a minor revision may be beneficial to further clarify certain points and enhance the overall impact of the work. For instance, a more detailed explanation of the implications of this work for the development of functional foods would be advantageous. Additionally, further discussion of the specific roles of the peptides identified would enhance the article's value to the broader scientific community.

In conclusion, this article offers a significant contribution to the field and with minor revision, it should be ready for publication. The authors have demonstrated excellent research, and their findings suggest promising avenues for future exploration in the field of functional foods and bioactive peptides.

Round 2

Reviewer 1 Report

At this form, it can be accepted for publication.